# Inter-continental Data Centre Power Load Balancing for Renewable Energy Maximisation

Rasoul Rahmani , Irene Moser * and Antonio L. Cricenti

Department of Computer Science and Software Engineering, Swinburne University of Technology, Melbourne, VIC 3122, Australia; rasoulrahmany@gmail.com (R.R.); tcricenti@swin.edu.au (A.L.C.)
* Correspondence: imoser@swin.edu.au

**Abstract:** The ever increasing popularity of Cloud and similar services pushes the demand for data centres, which have a high power consumption. In an attempt to increase the sustainability of the power generation, data centres have been fed by microgrids which include renewable generation—so-called 'green data centres'. However, the peak load of data centres often does not coincide with solar generation, because demand mostly peaks in the evening. Shifting power to data centres incurs transmission losses; shifting the data transmission has no such drawback. We demonstrate the effectivity of computational load shifting between data centres located in different time zones using a case study that balances demands between three data centres on three continents. This study contributes a method that exploits the opportunities provided by the varied timing of peak solar generation across the globe, transferring computation load to data centres that have sufficient renewable energy whenever possible. Our study shows that balancing computation loads between three green data centres on three continents can improve the use of renewables by up to 22%. Assuming the grid energy does not include renewables, this amounts to a 13% reduction in $CO_2$ emissions.

**Keywords:** data centre load balancing; green data centre; multi-agent systems



## 1. Introduction

Data centres have large and growing energy requirements and are becoming significant consumers of energy. The Green Data Centre (GDC) architecture is one of the technologies proposed to alleviate this problem [1–3] by reducing consumption and increasing energy efficiency as well as utilising energy from sustainable sources [4,5]. Renewable power generation systems require comprehensive power control and energy management methods which can cope with the intermittent nature of these resources. Such technologies have recently been devised for microgrids [6–8].

One of the challenges of renewable energy is how to make use of excess generation. Excess solar power is usually either stored in batteries or fuel cells or fed back into the main grid. Energy storage systems are still expensive and require complex management and monitoring systems. If a solar farm uses battery storage, capital investment, operational and maintenance contribute significantly to the net present cost or the project life-cycle cost [9,10]. Typically solar generation does not coincide with peak demand in the main grid, which typically occurs during the evening hours. This means that solar power generation without comprehensive power management and storage does not reduce the use of grid power to its potential. Many governments are now providing incentives for solar power generation due to environmental aspects related to greenhouse gas emissions and global warming [11,12]. The distance between the power generation facility and the location of the load is also an important consideration, as power loss increases with increasing transmission distances, resulting in additional costs [7,13].

Closely related to GDC research are studies on the geographical load balancing (GLB) of cloud systems [14–17], which discuss scheduling and server utilisation approaches

designed to save energy. Other approaches introduce detailed power modelling and intelligent management of network and servers [18]. Energy management and optimisation of green energy in distributed data centres [19,20] have also been researched. Overall, the existing literature focuses on the management of GDCs which are connected by means of dedicated high-bandwidth communication links while being served by the same power system. Existing work tends to assume GDCs to be in the same country and time zone, leading to synchronous peak loads, which are caused by high processing demand observed predominantly in the late evening [21,22].

The algorithms applied in existing GLB methods try to perform a global yet centralised control and management strategy towards obtaining optimal load balancing or energy management—most of this work relates to to data centres located in the United States. Such centralised systems suffer from several drawbacks, such as a large computational cost born by the unit responsible for the optimisation, the need for access to local information of all data centres, i.e., data confidentiality, privacy and security concerns, as well as low resilience due to centralisation [23,24].

In this paper, we propose a new agent-based approach to maximise the utilisation of sustainable power generation by effectively performing "electrical power transmission over internet cables" by shifting a GDC's computational load. The goal is to integrate the existing structures of green data centres which are intercontinentally dispersed. The fact that different time zones have different sustainable power peak generation is the key to manage integrated data centres by optimally transferring data among them which results in electrical power balancing with the goal of maximising sustainable energy consumption.

In Section 2 of this paper, we present a comprehensive power model for a data centre and a renewable energy based microgrid, by considering the interactions between their internal components and the environmental impacts. In addition, a model based on the power consumption is provided for the data transfer between data centres, followed by an agent-based management of the microgrid containing GDC. Furthermore, a case study consisting of three intercontinentally dispersed GDCs (located in Australia, Switzerland and California) with time-zone differences is presented to analyse the performance of the proposed method. In Section 3, the objective function for an integrated system consisting of $\mathcal{N}$ microgrids is presented to ensure optimal operation of the whole system with the objective of maximising sustainable energy consumption. In Section 4, the case study and simulation parameters are discussed in addition to explaining the Base Scenario of the simulation. Section 5 is dedicated to the obtained results and discussion based on the case study presented. Finally, Section 6 presents the major findings of the research.

## 2. System Model

*2.1. Data Centre Power Management Model*

The load balancing model introduced here considers aspects of power management in GDCs developed in a separate publication [21]. In the following, we repeat the aspects of the power management model that are relevant for an understanding of the current work.

To be able to make good decisions when to offload data processing requests to other centres, we modelled the power consumption of the information technology equipment (ITE), cooling system and auxiliary systems such as lighting, security, control and monitoring which we combine under miscellaneous power consumption.

If the server cluster is assumed to be homogeneous and the load balancing is perfect, the usage of each server in the server farm is the same. For simplicity, the current work assumes this is the case. Modelling the individual loads of servers is possible [25], but would not affect the outcome.

The power consumption of the servers for a given load can be modelled as a linear relationship as shown in Equation (1);

$$\mathcal{P}_{sf} = \sum_{i=1}^{\mathcal{N}} p_i^{idle} + [(p_i^{peak} - p_i^{idle}) \times \mathcal{U}] \tag{1}$$

where $\mathcal{N}$ represents the number of servers, $\mathcal{U}$ the utilisation of the servers, and $p_i$ the power consumption of the $i$th server, while $p_i^{idle} = 120$ W and $p_i^{peak} = 250$ W are constants.

The uninterruptible power supply (UPS) system in a data centre is a battery backup that provides enough time to properly power down the equipment when the power supply fails. The UPS system is distinct from the energy storage system which is charged/discharged as part of the power management strategies of the model. The UPS supplies power to the power distribution units (PDUs), which dispatch the electricity to the server racks. The power consumption of the UPS and PDU units are calculated based on the servers' power consumption and their sum is considered as total supply power consumption ($\mathcal{P}_{sup}$). Hence, the total power consumption of the ITE can be expressed as in Equation (2).

$$\mathcal{P}_{ITE} = \mathcal{P}_{sf} + \mathcal{P}_{sup} \tag{2}$$

The cooling systems of data centres vary based on the technologies involved and equipment used. Details about modelling the popular cooling systems of data centres is presented in [21]. However, the parameters affecting the power consumption of this system regardless of its structure are mainly the power consumption of the servers, which is a function of their utilisation $\mathcal{U}$, and the ambient temperature $T$ of the data centre.

In this paper, a very popular cooling system structure is taken into account for the simulation purpose which consists of CRAH units, a chiller plant, and water pumps.

The power consumption of the CRAH unit $\mathcal{P}_{CRAH}$ and the chiller plant $\mathcal{P}_{chiller}$ are usually affected most by the difference between the water temperature before and after cooling the computer room, which is related to the heat generated by the servers. As the chiller plant is located outside of the computer room, the ambient temperature plays role in the power consumed for transferring heat from indoors to outdoors.

Equation (3) shows the energy efficiency ratio (EER) of a cooling system plant as a function of the ambient temperature [26]. To calculate the effect of the ambient temperature on the power consumption of the cooling system, the power consumption must be divided by $\mathcal{F}_{EER}$.

$$\mathcal{F}_{EER}(T) = -10^{-4} \times T^2 - 0.0726 \times T + 5.8451 \tag{3}$$

where $T$ is the ambient temperature of the data centre in $°C$.

Based on the models developed, the total power consumption of the cooling system in a data centre ($\mathcal{P}_{COOL}$) can be obtained using Equation (4).

$$\mathcal{P}_{COOL} = \frac{\mathcal{P}_{chiller} + \mathcal{P}_{CRAH} + \mathcal{P}_{pump}}{\mathcal{F}_{EER}(T)} \tag{4}$$

where $\mathcal{P}_{pump}$ denotes the power consumption of the pumps used in the cooling system.

The total power consumption of this category ($\mathcal{P}_{MISC}$) is considered to be 6% of peak demand of the data centre [27].

The total power load of a data centre ($\mathcal{P}_{DC}$) is the summation of its components (5).

$$\mathcal{P}_{DC} = \mathcal{P}_{ITE} + \mathcal{P}_{COOL} + \mathcal{P}_{MISC} \tag{5}$$

## 2.2. Microgrid Power Generation Model

The generation of photovoltaic (PV) panels mainly depends on the solar irradiation and ambient temperature which can be obtained from Equation (6), which has first been published by Logenthiran and Srinivasan [28].

$$\mathcal{P}_{PV} = \frac{\mathcal{H}}{1000} \times \left[ \mathcal{P}_{max} + \mu_{P_{max}} \left( \mathcal{T}_{amb} + \mathcal{H} \frac{NOCT - 20}{800} - 25 \right) \right] \tag{6}$$

where $\mathcal{H}$ is the solar irradiation in W/m$^2$, $\mathcal{P}_{max}$ is the peak power being generated by the PV panel, $\mu_{P_{max}}$ denotes the temperature coefficient of the maximum power point, $\mathcal{T}_{amb}$ is

the ambient temperature of the PV panel, and *NOCT* denotes the normal operating cell temperature of the PV panel.

The power generation model used for the wind turbines with a power rating of 100 kW is shown in Equation (7).

$$\mathcal{P}_{WT} = 1.43 \times \mathcal{V}^2 - 4.29 \times \mathcal{V} \tag{7}$$

where $3 \leq \mathcal{V} \leq 25$ is the wind speed in m/s, and $\mathcal{P}_{WT}$ is equal to zero for other values of the wind speed due to cut in and cut off constraints of the wind turbine.

The total renewable power generation of the microgrid ($\mathcal{P}_{RE}$) is obtained using Equation (8).

$$\mathcal{P}_{RE} = \mathcal{P}_{PV} + \mathcal{P}_{WT} \tag{8}$$

In energy management systems, usually the mathematical model of the battery system includes the state of charge (SOC) and charging/discharging rate (C-rate) [28] while detailed PDE modelling for the voltage level and its associated constraints as presented by [29], is neglected. In general, the charging and discharging times versus C-rate depend on the battery type. Although a high C-rate leads to fast charging and discharging of the battery [30], at the same time, it degrades the battery life-time and efficiency faster. Regarding SOC of the battery system, some constraints have to be taken into account. As suggested by [31], a SOC of greater than 20% can help to bridge the gap between the renewable energy shortage and the load demand. On the other hand, overcharging the battery to more than 80% of SOC is not recommended because it may cause physical damage to the storage system.

### 2.3. Data Transfer Model

In general, bulk data is transferred across data centres through "highways", parts of the internet core network architecture [32,33], which can carry a traffic from 10 Gbps to 100 Gbps [34]. In 2014, a new technology of data transfer using single-mode fibre, which is able to carry an internet traffic of 255 Tbps [35], was tested successfully. Although the new technology has not been commercialised yet, in the near future, a commercial bandwidth of 1 Tbps does seem realistic. If the data is transferred between two data centres, the following consequences are to be expected:

1. Reduction in the power load of sender data centre;
2. Increase in the power consumption of the receiver data centre;
3. Energy consumed at the core routers for transferring the data as packets.

The change in power consumption of a data centre respective to the change in data size, can be calculated by converting the data size into change in the server utilisation ($\Delta \mathcal{U}$), as shown in (9),

$$\Delta \mathcal{U} = \frac{\mathcal{D}_t \times \mathcal{N}_{spr}}{\mathcal{D}_r \times \mathcal{N}_s} \tag{9}$$

where $\mathcal{D}_t$ is the transfer data size in Bytes, $\mathcal{D}_r$ is the size of data in Bytes that can be stored in each server rack, $\mathcal{N}_{spr}$ denotes the number of servers per rack, and $\mathcal{N}_s$ is the total number of servers in the data centre. It is assumed that the use of each server has a linear relationship with the data size. Based on the power consumption model of a data centre presented in Section 2.1, we can calculate the change in power consumption of each data centre by knowing the change in its server utilisation.

A recent study by [36] investigates the power consumption of Future Internet Architecture (FIA) and IP networks in detail, using different storage technologies. They assume that the total power consumption of a core router consists of the items baseline power consumption, forwarding and decision making, and content catching. Each item has a rating based on the bandwidth. In the current study, we assume that the power consumption is linear in the bandwidth occupied by the data being transferred with a constant power rating. As an example, if we occupy 100 Gbps of a core router CRS-3 which has a maximum power of 7.66 kW with 24 slots 400 Gbps (a throughput of 9.6 Tbps), the power consumption of the router due to our bandwidth will be 79.8 W [37]. In practice, this value should be smaller

due to the fact that there is a fixed idle power consumption for any core router which is irrelevant to the size of bandwidth and data transfer. Therefore, the power consumption due to data transfer ($\mathcal{P}_T$) is roughly estimated by Equation (10),

$$\mathcal{P}_T = \sum_{i=1}^{\mathcal{N}_{cr}} \mathcal{P}_{cr}^i(BW), \qquad (10)$$

where $i$ denotes the number of core routers, $\mathcal{N}_{cr}$ denotes the total number of the core routers between the origin and destination, and $\mathcal{P}_{cr}$ is the power consumption of the $i$th core router as a function of bandwidth occupied $BW$.

*2.4. Agent-Based Management Architecture*

The agent-based management algorithm used in this research is a modification of the approach published by [23].

Based on the control and management architecture shown in Figure 1, the agents have the following roles:

- Microgrid Controller Agent (MGCA) is mainly responsible for ensuring the optimal operation of the microgrid to maintain the power balance and other constraints. It has the ability to communicate with all agents dynamically and perform optimisation calculations. It also informs all agents about the optimal operation at the end of negotiations with external microgrid agents.
- Green Data Centre Agent (GDCA) represents the agent which controls and monitors the electrical power and data load balancing within the green data centre by communicating with sub-agents located inside the GDC. It also communicates with the MGCA for sending and receiving control and management information.
- Sustainable Energy Agent (SEA) monitors the generation of sustainable energy (PV and wind) and updates the MGCA with this information at each time step considered. It is physically connected to the generation unit, in order to relay the commands from the MGCA in regards to power management.
- Storage Agent (SA) communicates with the MGCA and is physically connected to the storage units. The SA controls the charging and discharging of the storage electricity on demand. It can keep track of different predefined status values for delivery to the MGCA, if required.
- Grid Agent (GA) switches the connection between the microgrid and the main grid upon request. It receives commands from the MGCA regarding the optimal system operation. It also updates the MGCA with information about the electricity price of the main grid.
- Forecast Agent (FA) is responsible for providing the MGCA with forecast data regarding the generation of sustainable energy. The information provided by the FA is used by the MGCA in the negotiations with other MGCAs in an integrated system.

While the MGCA is responsible for ensuring the power balance and optimal operation of the microgrid, it also negotiates with other MGCAs in an integrated system using a negotiation platform in order to increase the sustainability of the integrated microgrids. There exist feasible and practical negotiation methods in the literature [38,39]; however, this research is not focusing on the negotiation methods and their quality. Instead its focus is on the feasibility of negotiation on the electrical energy by transferring data between the green data centres located in different microgrids.

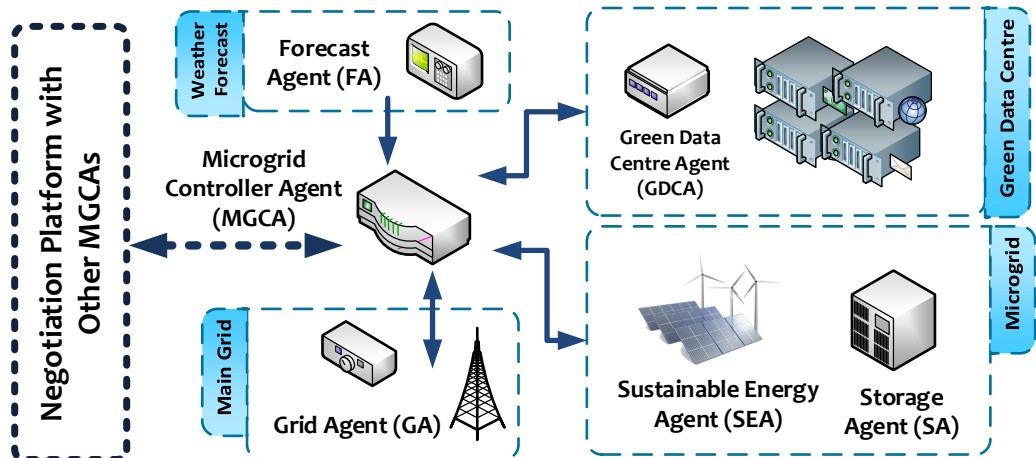

**Figure 1.** The structure of the agent-based management of the microgrid including the green data centre.

## 3. Optimising Sustainability

The main goal of this research is to demonstrate how the use of sustainable energy in intercontinentally dispersed green data centres fed by microgrids with renewable generation can be optimised through computational load balancing. Data centres that have insufficient renewable energy for their processing requirements can 'offload' tasks to other data centres, but the system must meet its service level agreements (SLAs), meaning the data transfer time must not delay the response beyond acceptable limits.

The data centre management and reporting is carried out on an hourly basis, as illustrated in Figure 2. First, the MGCA receives forecast weather data from the FA, the server utilisation from the GDCA, and the state of charge of batteries from the SA. Then it estimates the GDC load demand and renewable generation using the total power consumption model of the data centre and renewable power generation models. Taking into account the estimated power generation, load demand, and available battery capacity, the excess renewable energy is estimated. Based on this data, the MGCA computes the near-optimal quantities of computation loads to exchange as follows.

For the $i$th microgrid, we denote the excess renewable generation as $\delta \mathcal{P}_G^i$. By transferring a certain size of data from data centre $j$ to $i$ (denoted as $\mathcal{D}_t^{ij}$), the data centre power load of the $i$th microgrid increases by $\Delta \mathcal{P}_{DC}^{ij}$. Another constraint is that any data centre should only accept processing loads which do not exceed its own capacity. This results in the optimisation problem in Equation (11):

$$min\left(\sum_{ij} \frac{\Delta \mathcal{P}_{DC}^{ij} + \mathcal{E}_T^{ij}}{\mathcal{D}_t^{ij}}\right) \tag{11}$$

$$subject\ to \quad 0 \le \delta \mathcal{P}_G^i - \sum_{ij}\left(\Delta \mathcal{P}_{DC}^{ij} + \mathcal{E}_T^{ij}\right),$$

$$0 \le 1 - \left(\mathcal{U}^i + \sum_{ij} \Delta \mathcal{U}^{ij}\right),$$

where $\mathcal{E}_T^{ij}$ represents the energy consumed by the data transfer from data centre $j$ to $i$, and depends largely on the number of routers on the way and the amount of the data, $\mathcal{U}^i$ is the utilisation of $i$th data centre and $\Delta \mathcal{U}^{ij}$ is the change in its utilisation due to transferring processing load from $j$th data centre to it.

To solve this optimisation problem, we used the Radial Movement Optimisation (RMO) technique which is a swarm-based metaheuristic algorithm proposed by Rahmani et al. [40,41], which has been shown to be effective at producing solutions of good quality to similar

problems. However, any other heuristic, for example particle swarm optimisation or differential evolution could also be applied to the model proposed here. Radial Movement Optimisation uses a vector-based search space to model the problem environment where each particle at each step proposes a solution to the problem. The particles must not exceed the boundaries of the search space, and the search space must include all possible solutions to the problem [42,43].

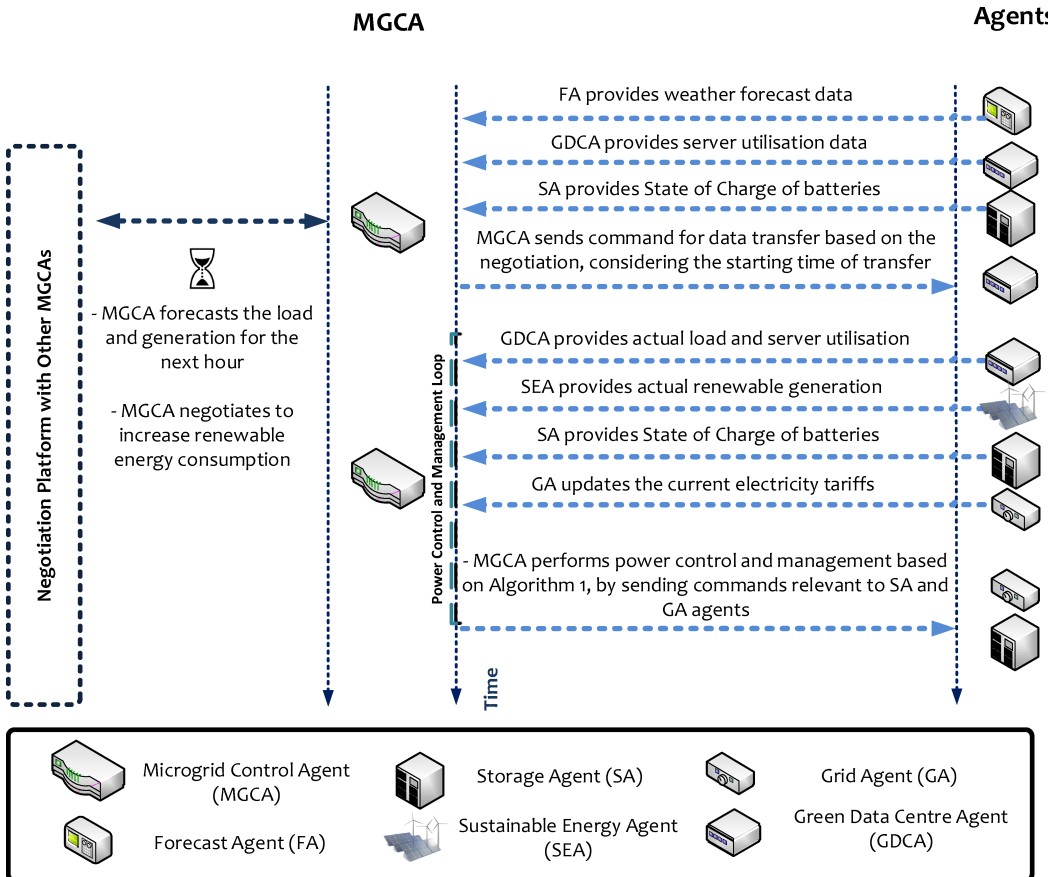

**Figure 2.** Schematic diagram of the hourly communications between the agents.

A data centre only dispatches processing loads which it cannot cover by sustainable energy. Once the optimised transfer loads have been established, the MCGA communicates them to the other MGCAs with the goal of increasing the overall sustainable energy usage in the integrated system. In our results, the outcomes of the optimisation process are applied, but in reality, it would be possible for a negotiation process to take place. The MGCA sends the data transfer commands to the GDCA that includes the starting time of transfer based on the negotiation results. The power control and management loop is carried out throughout the hour and manages the RE produced or shortfall incurred as presented in Algorithm 1.

---

**Algorithm 1:** Management of RE generation excess or shortfall.

---

1   $\delta P_{RE} \leftarrow P_{RE}^{actual} - P_{GDC}^{actual}$ // difference between generation and load;
2   **if** $\delta P_{RE} \leq 0$ **then** // checking for shortage
3     $\delta E_{SS} \leftarrow E_{SS}^{actual} - E_{SS}^{min}$ // calculating available stored energy;
4     **if** $\delta E_{SS} > 0$ **then**
5       **if** $\delta E_{SS} \geq P_{GDC}^{actual}$ **then** // sufficient battery charge
6         $P_G \leftarrow 0$ // no purchase from grid;
7         $E_{SS}^{actual} \leftarrow E_{SS}^{actual} - \delta E_{SS}$ // battery discharge;
8       **else**
9         $P_G \leftarrow (P_{GDC}^{actual} - P_{SE}^{actual}) - \delta E_{SS}$// purchase from grid;
10       **end**
11     **end**
12 **else**
13     // there is at least sufficient RE
14     $\delta E_{SS}^{cap} \leftarrow E_{SS}^{max} - E_{SS}^{actual}$ // calculating available storage capacity;
15     **if** $\delta E_{SS}^{cap} > 0$ **then** // storage capacity is available
16       **if** $\delta P_{RE} \geq \delta E_{SS}^{cap}$ **then** // more generation than storage capacity
17         $E_{SS}^{actual} \leftarrow E_{SS}^{max}$ // storage fully charged;
18         $P_G \leftarrow \delta P_{RE} - \delta E_{SS}^{cap}$ // excess fed into grid;
19       **else**
20         $E_{SS}^{actual} \leftarrow E_{SS}^{actual} + \delta P$// excess is all stored;
21       **end**
22     **else**
23       $P_G \leftarrow \delta P_{RE}$ // all goes to grid;
24     **end**
25 **end**

---

## 4. Evaluation

This section describes the experimentation conducted to demonstrate the effectiveness of redistributing computational loads for the purpose of maximising the use of renewable energy. The GDCs are expected to be geographically dispersed so that their peak renewable generation happens at different times of the day or even year. A suitable case study has been devised to create realistic scenarios for a comparison with a base scenario that does not rebalance computation loads. The optimisation procedure and settings are also explained in this section.

### 4.1. Case Study

As a case study, we consider three GDCs. The first is assumed to be located in Perth (named GDC1), Western Australia, the second in San Francisco, California, USA (named GDC2), and the third in Bern (named GDC3), Switzerland. The reasons for choosing these cities is that they cover a wide range of weather patterns which happen at different times of day due to the longitudinal differences of the locations around the globe. Choices from both the northern and southern hemispheres lead to seasonal conditions complementing each other between the data centres. Each microgrid is comprised of a PV array with maximum capacity of 20 MW, 100 wind turbines as defined by Equation (7), and a battery storage system with a capacity of 4 MWh and the maximum charging/discharging power of 600 kW. The SOC of the battery systems is kept between 20% and 80% in the simulation.

There is a 16 h time difference between Perth and San Francisco, which means midday in Perth coincides with 8 p.m. (the previous day) in San Francisco and 8 p.m. in Perth coincides with 4 a.m. in San Francisco. Depending on daylight saving, Bern's time zone is 6–8 h from Perth's and 8–10 h from San Francisco. The locations for the GDCs were chosen such that there is solar generation in one location when it is night at the other. The difference

in time zones also means that the evening peak in internet use in San Francisco can be alleviated by diverting some of the load to a Perth-based DC. Placing GCDs in different hemispheres lets us explore seasonal opportunities for offloading of computation.

A 24-day simple moving average (SMA) of meteorological data consisting of ambient temperature, solar irradiation, and wind speed in the three regions is shown in Figure 3. The middle graph depicting solar irradiation illustrates seasonal differences between the northern and southern hemispheres particularly well, with the red (San Francisco) and green (Bern) lines rising in the middle of the year while the blue line (Perth) slumps at that time. We can also observe that the overall volume of renewable power generation in Bern is significantly smaller, caused by an average solar irradiation of 65.38 W/m$^2$ and a wind speed of 1.41 m/s while these values are 221.18 W/m$^2$ and 5.09 m/s for Perth. In San Francisco, the average annual solar irradiation and wind speed are 195.66 W/m$^2$ and 4.23 m/s which are both slightly less than that of Perth. Similar solar irradiations in Perth and San Francisco are explained by similar latitudes (37.7749° N and 31.9505° S, respectively), as is the significantly weaker irradiation in Bern, which has a latitude of 46.9480° N.

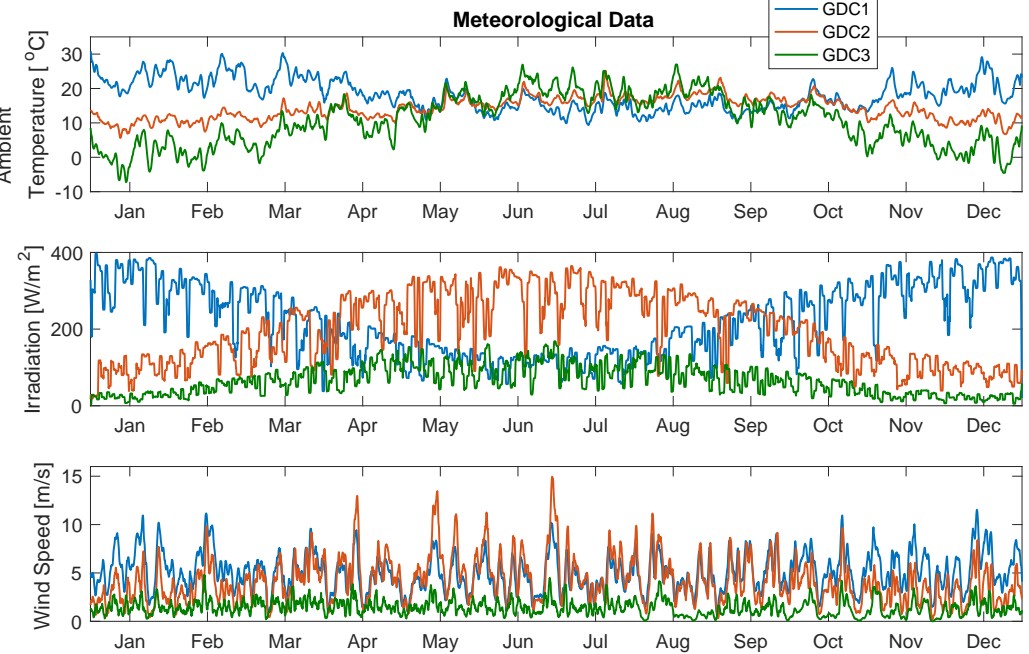

**Figure 3.** 24-day simple moving average of meteorological data for the three regions in the case study.

Considering the low average values of solar irradiation and wind speed for Bern compared with the other two cities, its renewable power generation is expectedly lower. The top graph in Figure 4 shows a 24-h SMA of the renewable power generation at each location. During the course of a year, the same microgrid in each of the three cities has renewable energy generation capacity of $6.042 \times 10^7$ kWh in Perth, $5.173 \times 10^7$ kWh in San Francisco, and only $1.031 \times 10^7$ kWh in Bern. While the annual renewable energy generation is close to each other for Perth and San Francisco, Bern can only generate about a sixth of Perth. The GDC server utilisation demand is assumed to be the same in the three regions, albeit offset by the time zone difference: The evening peak load, for example, is experienced when it is evening at the data centre under consideration. Ignoring the effects of daylight saving, the middle and lower graph illustrate the effects of Perth being 16 h ahead of San Francisco and 7 h ahead of Bern. The peak power load demand for each GDC is $2.150 \times 10^4$ kW while its average is $1.231 \times 10^4$ kW. The annual energy consumption of each data centre is $1.077 \times 10^8$ kWh. The average and annual renewable energy generation cannot support the total load demand on its own, *e.g.*, the annual renewable energy generation in Perth if fully utilised can cover about 56% of the

load demand. This has been a conscious design decision, as we wanted to investigate how capable our methodology is to utilise any amount of renewable energy which is unused and has to be fed in to the main grid, in a regular case. In other words, we did not oversize the microgrid in order to have a lot excess in the renewable energy generation. The portion of RE generation and load demand for the three microgrids are better shown in Figure 5 in which one can observe that most of excess RE generation for Perth and San Francisco happens during their summer seasons.

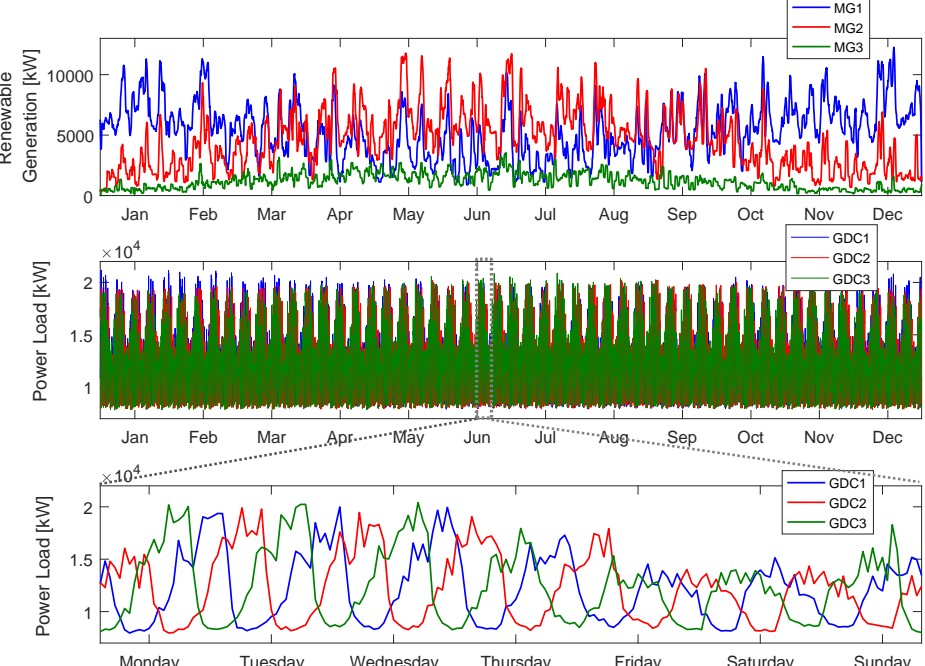

**Figure 4.** 24-day simple moving average of renewable power generation (top graph) and hourly GDC load demands for a year (middle graph) and a week (bottom graph) for the three microgrids in the case study.

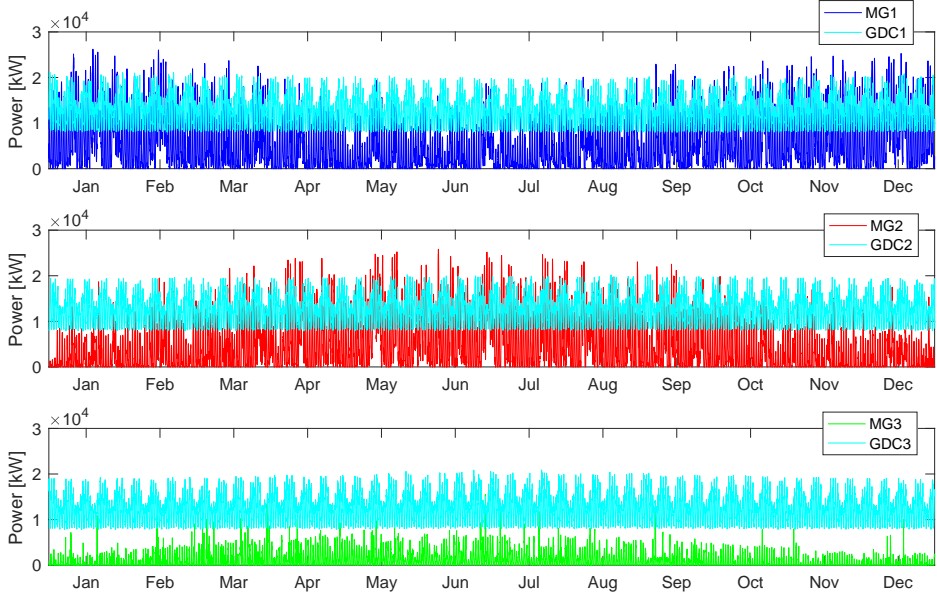

**Figure 5.** Hourly renewable power generation (MGx) and GDC load demand (GDCx) for the three microgrids in the Base Scenario. If the two lines reach the same height, the same amount of power was generated and used for computation load.

As shown in Figure 6, each GDC is supplied by a microgrid which is connected to the local main grid system and equipped with renewable power generation—PV panels and wind turbines. There is an energy storage system in each microgrid for storing the excess renewable energy generation. Each microgrid controller (MGC) has access to a local weather station to obtain forecast environmental data.

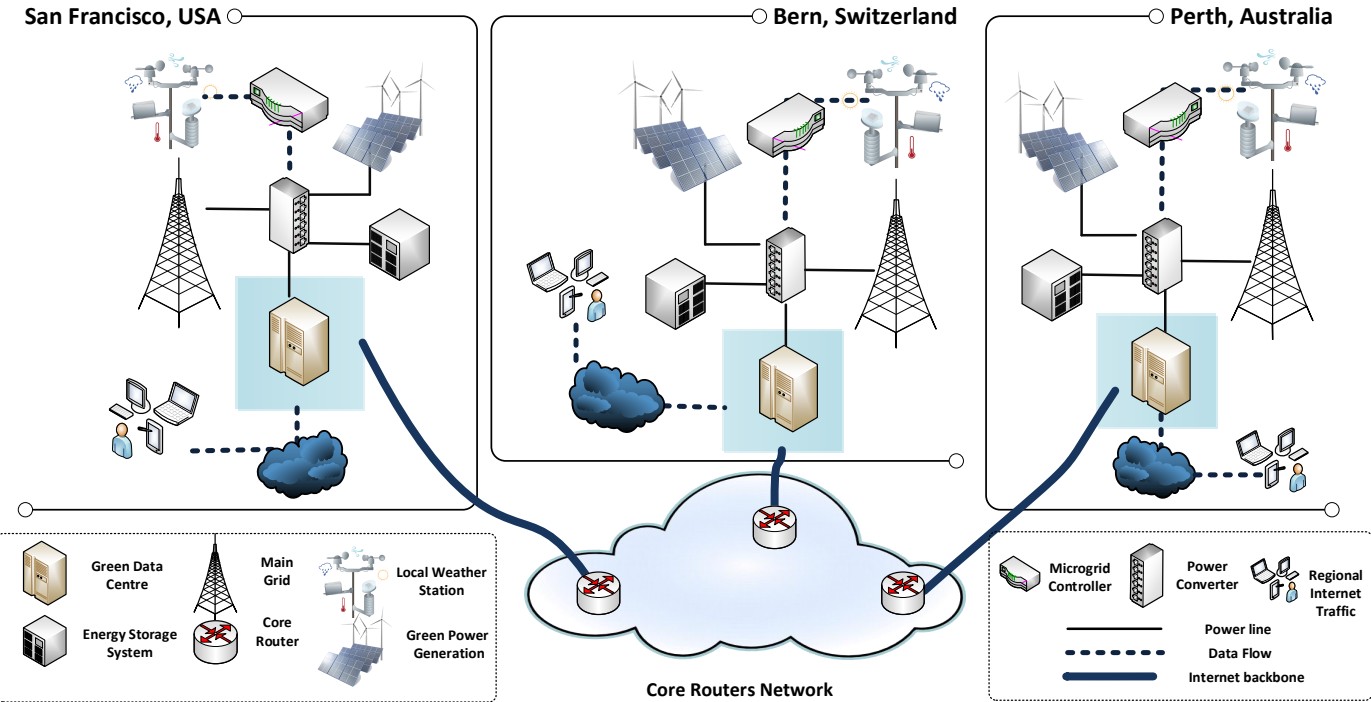

**Figure 6.** Case study of the three integrated green data centres located in Australia, California, and Switzerland.

### 4.2. Simulation

We compare three scenarios for the case study described in Section 4.1. The Base Scenario creates a baseline to compare against and assumes no data is transferred between the three GDCs. This is achieved by setting the maximal transfer allowance $\mathcal{D}_t^{max} = 0$. In Scenario 1, we consider a constraint of 100 TB per hour for the $\mathcal{D}_t^{max}$ based on the technology existing today (400 Gbps traffic), while Scenario 2 regards no constraint for the data transfer. The green data centres and their microgrids are identical for the three locations shown in Figure 6, which makes it easier to compare system performance. Each GDC contains 40,000 servers and its utilisation is assumed to be regional [21,22].

It is assumed that the internet traffic of each data centre is a priori local, meaning requests are sent to the local DC. Depending on the scenario, the local DC may offload some of the data to other centres. Regardless of scenario, each of the microgrids feeds its own data centre as described in Section 3 (no energy transfer takes place).

In our simulation, we establish the availability of solar and wind energy using historic hourly environmental data for both cities, comprising solar irradiation, wind speed and ambient temperature. This data was collected using the Meteonorm 7.1.11 software for the full year of 2016, which makes it possible to analyse the effects of different seasons of a year on the generation and consumption profiles. All algorithms were implemented in Matlab.

For the RMO algorithm, the number of particles is selected as 20 with a maximum of 30 iterations. The coefficients of RMO are set based on the findings of [40]. The search space consists of three dimensions in the range $[-\mathcal{D}_t^{max}, \mathcal{D}_t^{max}]$. The first dimension returns the data transfer size from GDC1 to GDC2, the second returns the data transfer size from GDC1 to GDC3, while the last dimension represents the data transfer size from GDC2 to GDC3. The value obtained from each dimension shows the size of data transfer at its direction at

the same time. For instance, if the value returned from the first dimension is 24, it means 24 TB from GDC1 to GDC2, while if the value from the same dimension is −38, it means that 38 TB to be transferred from GDC2 to GDC1. Each particle in the search space returns a set of data transfer between the three GDCs which will be evaluated using Equation (11). After each iteration, the RMO parameters will be updated based on the evaluated proposals from the particles, and then the next iteration starts.

## 5. Results and Discussion

The data set of computation loads compared to renewable power generation for all three GDCs over the full year that represents our simulation period is illustrated in Figure 5. The light blue lines labelled "GDCx" show the energy required for the computation in each DC while the blue, red and green lines, labelled "MGx" show the renewable generation of the microgrids. We can see that the renewable generation exceeds the energy needs of GDC1 (Perth) many times in September to February, while GDC2 (San Francisco) has excess renewable energy predominantly in April–July, and Bern's microgrid rarely produces sufficient renewables for its own GDC. Therefore the expectation is to see computation load shifts from Bern to San Francisco and Perth over the entire year, shifts from San Francisco to Perth in the period September to February and Perth to San Francisco in the summer of the northern hemisphere.

Figure 7 summarises where computation loads were processed during the entire year considered in the simulation. The colours denote the GDC the workload 'belongs to', i.e., the GDC which received the processing request in the first place (and possibly passed it on). Therefore, blue areas of a column depict computation loads that belonged to GDC1 (Perth), red areas are proportions of GDC2's data (San Francisco) and green blocks are processing work sent a priori to GDC3 (Bern). Each of the coloured blocks add up to 100%.

The leftmost column marked "GDC1" denotes the loads computed in Perth, the other columns illustrate the loads processed at GDC2 and GDC3, respectively. Expectedly, the vast majority of data computed at GDC1 was Perth's 'own' (the blue part), but Perth also processed loads originally sent to San Francisco and Bern. In Scenario 1, Perth processed 94.59% of its own data, 5.38% of Perth's data was processed in San Francisco (GDC2) and an insignificant remainder in Bern (GDC3). Bern almost exclusively processed its own data, offloading 8.65% to Perth and 6.65% to San Francisco over the year.

While Scenario 1 shows significant shifts of load, Scenario 2, which assumes no limits in processing power, demonstrates how much of each DC's load would be shifted to other DC's if the decision was purely made in the interest of using as much renewable power as possible. In our case studies, we assumed the microgrids and their renewable generation to be identical in all locations. Scenario 2 raises the question whether Bern's weather patterns would justify not having a DC at all and instead offloading their processing to areas better suited for renewable generation.

The annual breakdown of energy supply for each microgrid in different scenarios is shown in Figure 8. The amount of energy purchased from the grid to support each GDC has decreased for all microgrids in Scenario 1 and even more in Scenario 2, which does not impose a limit on the data transfers. In MG1, the share of renewable energy in supplying load demand becomes higher than the grid purchase in Scenario 2. The biggest change in grid purchase is shown in MG3, where the grid energy purchase has decreased from $9.53 \times 10^7$ kWh in the Base Scenario to $8.98 \times 10^7$ kWh and $8.32 \times 10^7$ kWh in Scenario 1 and Scenario 2, respectively. That is explained by the high amount of data transfer from GDC3 to GDC1 and GDC2 as shown in Figure 7.

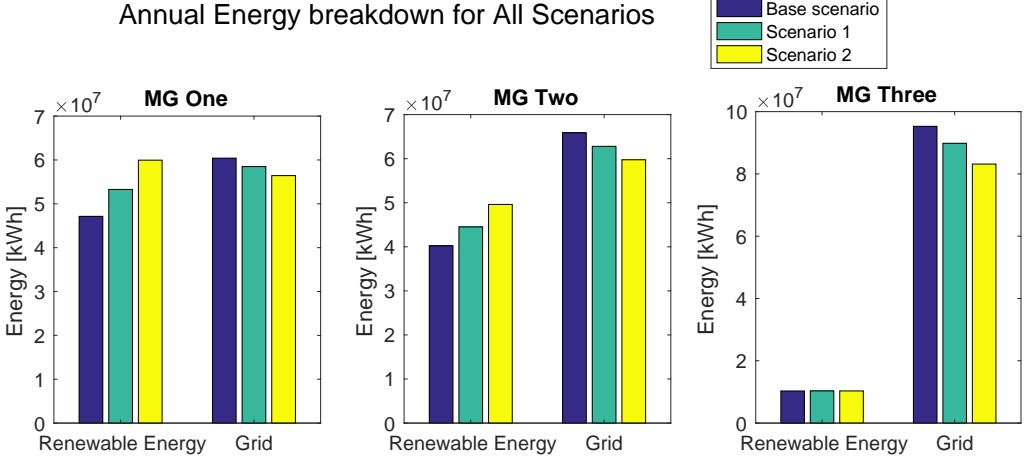

**Figure 7.** Total annual computational load of the three green data centres against the base scenario for (**a**) Scenario 1—limited data transfer, and (**b**) Scenario 2—unlimited data transfer, between the GDCs.

**Figure 8.** Annual energy breakdown for each scenario.

Figure 9 shows the percentage of the renewable energy generation actually used for data processing in the GDCs for the three scenarios. In MG1, only about 78% of the renewable energy generation was used by the GDC in the Base Scenario and the rest ($1.329 \times 10^7$ kWh) was fed into the main grid. Data load balancing was able to increase this value by 10.2 percentage points in Scenario 1 and 21.2 percentage points in Scenario 2 reaching 99.23% renewable energy usage. In MG2, the use of renewables reached 86.1% and 95.9% for Scenario 1 and Scenario 2, respectively, up from 78.8% in the Base Scenario. The annual renewable energy utilisation of MG3 is 100% for all scenario as Figure 5 suggests. MG3 has relatively low renewable energy generation, and all of it can be used for the data processing of the GDC3 load demand.

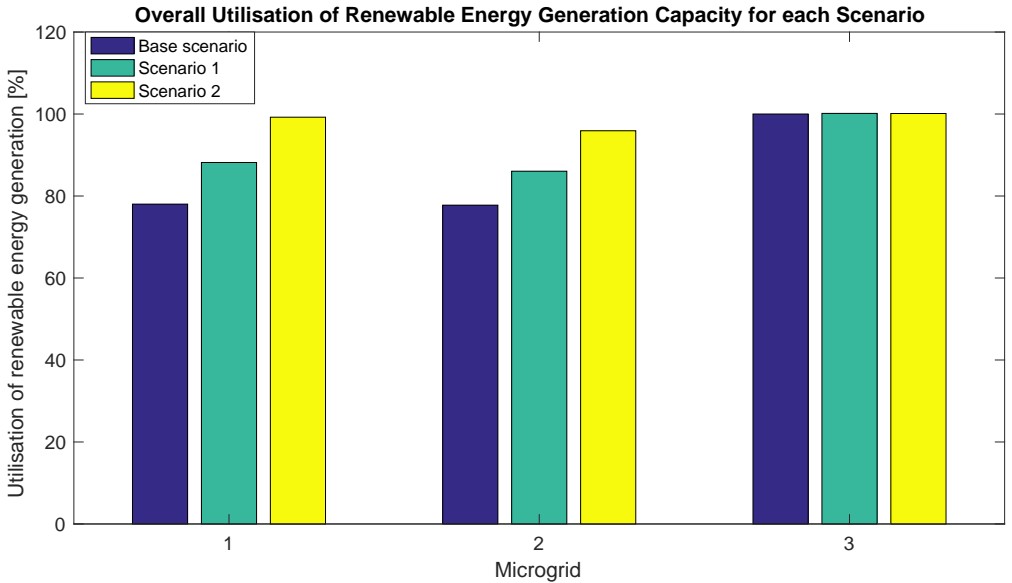

**Figure 9.** Percentages of renewable energy generation used by the GDCs. The remaining percentages are fed back into the grid.

If we consider the $CO_2$ emissions caused by the main grid, which is assumed to be a non-renewable power generation, as 850g per kWh electricity generation [44], the total $CO_2$ emissions for the three simulation conditions and their respective energy conservations are shown in Figure 10. It can be observed that the fraction of renewable energy in the annual energy supply breakdown for the three microgrids together has increased from 31% in Base Scenario ($1.002 \times 10^8$ kWh) to 34% and 38% for Scenario 1 and Scenario 2, respectively. This means that the data load balancing technique proposed in this work has increased the utilisation of renewable energy by $9.70 \times 10^6$ kWh in Scenario 1 and $2.25 \times 10^7$ in Scenario 2 without any change in the existing infrastructure of the microgrids. The biggest decrease in the $CO_2$ emissions can be observed in MG3 where the annual $CO_2$ emissions have decreased by 6% in Scenario 1 and 13% in Scenario 2 from $8.10 \times 10^7$ kg in Base Scenario.

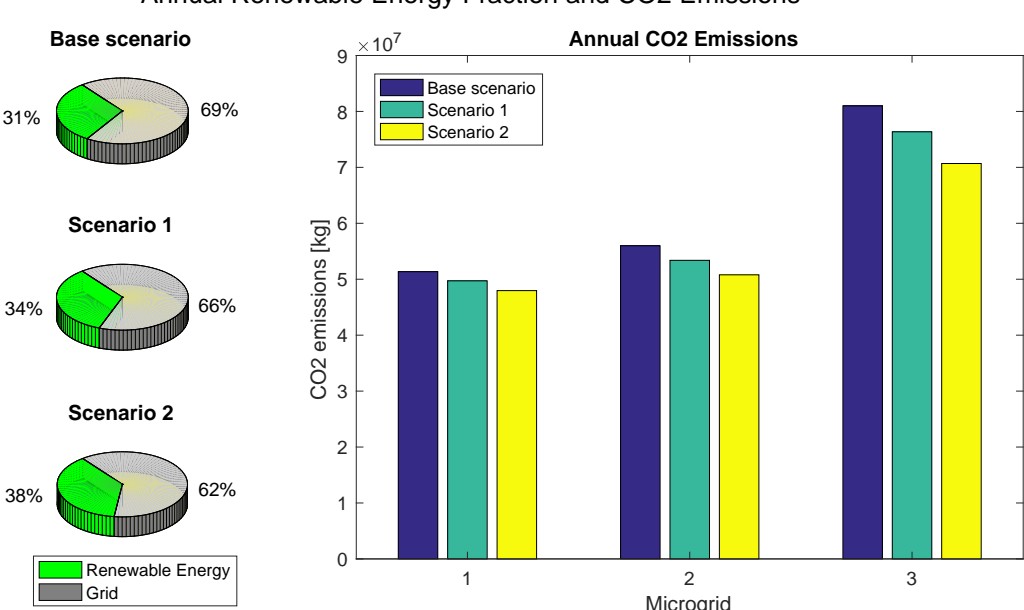

**Figure 10.** Annual renewable energy fraction and $CO_2$ emissions for each scenario.

## 6. Conclusions

The increasing computation loads of data centres internationally are an incentive to find new ways of sourcing cheap and sustainable energy for them. Existing studies have explored GDCs that share computation loads while relying on the same power sources. No previous study has explored the transfer of computation loads among unconnected GDCs in different time zones to match peak RE generation with peak computation loads. Renewable generation often peaks at a time of the day and season, with peaks for photovoltaics occurring predominantly during midday and summer time. Peak loads, however, are observed in the early evening. This provides an opportunity to maximise the use of sustainable energy by balancing computation loads across hemispheres and continents. We investigated a yearly computation load assumed to be identical in a Western Australian, US west coast and European data centres, and scenarios of sharing these loads between the three data centres. The renewable generation is derived from historic weather patterns. We also proposed an agent-based management architecture to maximise the utilisation of sustainable power generation among the intercontinentally dispersed green data centres. When we capped the maximal transfer allowance per hour at 100 TB, a reasonable assumption given contemporary Internet capacities, we observed an increase in the use of renewable energy by 9% and a decrease in $CO_2$ emissions by 6%. If we assume no limit, the increase in the use of renewables increases to 22% while decreasing emissions by 13%. Assuming identical capacities of renewable generation and computational load at all three locations, we observe that Western Australia produces the largest amount of renewable energy and imports the largest amount of computational load, computing 38% of the load of the European and 34% of the northern American data centres when no data transfer limits are imposed. We also see that a European data centre has hardly any renewable energy to spare for external loads. When no limits are imposed, it exports almost 70% of its load.

**Author Contributions:** Conceptualization, I.M. and R.R.; methodology, R.R. and I.M.; software, R.R.; validation, R.R., I.M. and A.L.C.; formal analysis, R.R., I.M. and A.L.C.; investigation, R.R.; resources, R.R.; data curation, R.R.; writing—original draft preparation, R.R.; writing—review and editing, I.M.; visualization, R.R.; supervision, I.M. and A.L.C.; project administration, I.M. All authors have read and agreed to the published version of the manuscript.

**Funding:** This research received no external funding.

**Conflicts of Interest:** The authors declare no conflict of interest.

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
