# Peer review of "Inter-continental Data Centre Power Load Balancing for Renewable Energy Maximisation"

_electronics, doi:10.3390/electronics11101564_

Round 1
Reviewer 1 Report
Review the Manuscript ID: electronics-1648174
Inter-continental data centre power load balancing for renewable energy maximisation
In this paper, the authors propose a new agent-based approach to maximise the utilisation of sustainable power generation by effectively performing "electrical power transmission over internet cables by shifting a Green Data Centre computational load. The goal is to integrate the existing structures of green data centres which are intercontinentally dispersed.
Authors should review the text of the article.
In section 2, there is an excessive use of subsections.
Section 3 is very summarized. Authors should explain in more detail the formulation of the optimization problem and the methodologies used.
In section 4.1, they mention that there are 100 wind turbines, it should be clear what the rated power is.
In the study, the authors consider that there is the same installed power in the three locations. The results presented allow us to identify that this is not the best choice. What is the authors opinion? and the reason for this choice.
Review the units used in the figures.
It is necessary to review the formatting of some references.
Reviewer 2 Report
The article deals with the optimisation of sustainable energy consumption in green data centres by means of partial transfer of computational load to centres located on other continents. In this sense, it does not seem reasonable to transfer power consumption to a generator located on another continent and at the same time provide for energy storage in each microgrid. The essence of renewable distributed generation is to bring the plants closer to the places of consumption in order to minimise losses in the transport and distribution of electrical energy. On the other hand, the following aspects should be considered:
- The model proposed for the power consumed by the servers (equation 1) does not seem coherent. The utilisation function of the servers is not expressed. The values considered as constants for pidle and ppeak are not adequately justified.
- The variation of load consumption in kW over the day and the differences for the three data centres in the case study (Figure 3) are not explained.
- It is redundant to consider in the system architecture UPS feeders for the GDCs and energy storage (Store Agent) for the microgrid according to figure 1.
- The units are not expressed for all quantities in the equations (e.g. equation 6 and 9).
- The disadvantages of data transfer time and power loss between green data centres separated on different continents over a large distance are not taken into account.
- No consideration is given to the fact that excess renewable energy produced by the microgrid near each data centre can be fed into the grid for local use. Considering energy storage (SA agent) implies an increase in costs.
- As the microgrids considered in section 4.1 are identical, it does not seem logical that the annual renewable energy production in Berne is approximately one-fifth of that in San Francisco and one-sixth of that in Perth. The difference in latitudes in the locations must be compensated by differences in the slopes for the PV installation.
- The numerical data given in section 6 only considers generation and consumption for the case study at dispersed locations without considering grid interconnection.
Reviewer 3 Report
The article presented is interesting - it deals with the problem of powering large data centres and optimising their operation in terms of electricity consumption. The authors consider the possibility of using renewable energy sources, noting the following problem - when energy consumption in a datacenter is the highest, renewable energy sources work with little or no power and try to solve this problem.
The structure of the article is in my opinion correct, introduction ok, I would suggest to highlight the contribution of the article more clearly.
The figures are clear and readable. Literature in the vast majority quite new.
My specific comments are as follows (they are very minor):
1) Please describe more extensively the optimization methods used.
2) In which software were the simulations performed? It seems to me that this information is missing from the body of the article.
Best regards to the Authors!
Reviewer 4 Report
The overall quality of this manuscript is good. The subject investigated in this paper is in the scope of the journal. However, both readability and presentation of the paper should be checked and improved.
Abstract should include the significance and importance of the work. It should also discuss more about the proposed method.
Motivation and objectives should be discussed in details. Recent studies from high impact factor journal (see https://www.scimagojr.com/) should be cited like from IEEE transactions, Springer and Elsevier in the introduction or in a related work section.
Conclusion: What are the advantages and disadvantages of this study compared to the existing studies in this area?
Reviewer 5 Report
There are some part I don't agree. For example - If a solar farm uses battery storage, capital investment, operational and maintenance contribute significantly to the net present cost or the project life-cycle cost. Because if we speak about production, operation and ecological waste management, it is not clean.
If I accept your premises, which are fairly reasonably listed, I have a few reservations about the conclusions that arose:
- how deep, or with what time granulation, meteorological data are processed?
- What is the prediction accuracy of your agent solution?
- do you assume that the theoretical use of a higher transmission speed will reduce the energy costs for data transmission? what about, for example, transmission infrastructure with the mentioned high level of transmission quality?
- You used 850 g per kWh to calculate emissions. This value according to the citation applies to China. However, each location has a different parameter, will it have a major impact on the computational model?
- the parameter mentioned above was for 2015, isn't it different?
- I recommend emphasizing what makes the article innovative
- Write the results of the simulations in points that describe the benefits of the above method
- What was the aim of the article, to prove the effectiveness of the solution? then it would be appropriate to compare the result with the value that will arise without this optimization
- from figures 5+ I don't see solution of mentioned problem of data processing at evening time, bring your optimization solution ?
Round 2
Reviewer 1 Report
Review the Manuscript ID: electronics-1648174
Inter-continental data centre power load balancing for renewable energy maximisation
In this paper, the authors propose a new agent-based approach to maximise the utilisation of sustainable power generation by effectively performing "electrical power transmission over internet cables by shifting a Green Data Centre computational load. The goal is to integrate the existing structures of green data centres which are intercontinentally dispersed.
The review/correction performed by the authors allowed an improvement in the quality of the paper.
Author Response
The reviewer appears satisfied with our modifications. The paper will be proofread again to remedy any linguistic problems.
Reviewer 2 Report
See comments in attached file.
Kind regards

Reviewer 5 Report
I think I can understand your comments and additional changes in article.
Author Response
The reviewer appears satisfied with the changes made.
Round 3
Reviewer 2 Report
With regard to the authors' last reply, I would like to make the following comments:
- The Dell PowerEdge server models referred to are no longer marketed. There is no textual indication of their technology or whether their electrical characteristics are extensible to other models. The linear model used may have a negligible error for the actual power consumption of a single server, however, the error will be of a greater magnitude when the number of servers is large.
- It does not seem reasonable that the authors use Dell routers to evaluate the linear behaviour of equation 1 and another unidentified model of Cisco routers to evaluate the power consumption and performance values as explained in section 2.3.
- It is not envisaged that the excess renewable energy produced by the microgrid of each CDG can be fed into the electricity grid for local use. It does not seem reasonable for excess energy to be used for computational load from other continents, as this transfer would entail energy losses.
- Solar irradiation data in W/m2 from meteorological databases are expressed on a horizontal surface. It is understood that the authors have considered the optimal tilt at each location for maximum PV power generation.
